# Biomechanical study of anterior and posterior pelvic rings using pedicle screw fixation for Tile C1 pelvic fractures: Finite element analysis

**Yuanzheng Song, Changsheng Shao, Ximing Yang [ID]\*, Feng Lin**

Department of Orthopaedics, Tengzhou Central People's Hospital Affiliated to Jining Medical University, Tengzhou, Shandong Province, China

\* yangximing1979@163.com

## Abstract

### Objective

The purpose of this study was to analyse the biomechanical characteristics of pedicle screws with different placement methods and diameters in the treatment of Tile C1 pelvic fractures by finite element simulation technology and to compare them with the plate fixation model to verify the effectiveness of pedicle screw fixation.

### Methods

A three-dimensional digital model of a normal pelvis was obtained using computed tomography images. A finite element model of a normal pelvis containing major ligaments was built and validated (Model 1). Based on the verified normal pelvis finite element model, a Tile C1 pelvic fracture model was established (Model 2), and then a plate fixation model (Model 3) and a pedicle screw fixation model with different screw placement methods and diameters were established (Models 4–15). For all pelvic fracture fixation models, a vertical load of 500 N was applied on the upper surface of the sacrum to test the displacement and stress distribution of the pelvis in the standing state with both legs.

### Results

The finite element simulation results showed the maximum displacement of Model 1 and Models 3–15 to be less than 1 mm. The overall maximum displacement of Models 4–15 was slightly larger than that of Model 3 (the maximum difference was $177.91 \times 10^{-3}$ mm), but the maximum displacement of iliac bone and internal fixation in Models 4–15 was smaller than that of Model 3. The overall maximum stress (maximum stress of the ilium) and maximum stress of internal fixation in Models 4–15 were less than those in Model 3. The maximum displacement difference and maximum stress difference at the fracture of the pubic ramus between each fixed model were less than 0.01 mm and 1 MPa, respectively. The greater the diameter and number of pedicle screws were, the smaller the maximum displacement and stress of the pelvic fracture models were.The maximum displacement and stress of the

**Data Availability Statement:** All relevant data are within the paper.

**Funding:** This study was supported by Jining Medical University (grant number

JYFC2019FKJ186). The funders had no role in study design, data collection and analysis, decision to publish, or preparation of the manuscript.

**Competing interests:** The authors have declared that no competing interests exist.

pelvic fracture models of the screws placed on the injured side of the pubic region were smaller than the screws on the healthy side.

## Conclusion

Both the anterior and posterior pelvic rings are fixed with a pedicle screw rod system for treatment of Tile C1 pelvic fractures, which can obtain sufficient biomechanical stability and can be used as a suitable alternative to other implants. The greater the diameter and number of pedicle screws were, the greater the pelvic stability was, and the greater was the stability of the screws placed on the injured side of the pubic region than the screws on the healthy side.

## Introduction

Pelvic fractures are mostly caused by high-energy injuries, such as falls from heights and traffic accidents, accounting for 3% to 8% of all body fractures [1]. Tile type C fractures are vertically rotationally unstable fractures and often occur with shock or organ damage; despite accounting for only 0.34% of all pelvic fractures [2], the mortality rate can reach 31% [3, 4]. Treatment of this type of pelvic fracture has always been a debated and difficult point in the field of trauma orthopaedics. Due to destruction of the anterior and posterior pelvic ring structures with this type of fracture, the pelvis is asymmetric and unstable, and simple conservative treatment usually does not achieve good results. Most scholars recommend early surgery to restore stability of the pelvic ring, promote functional recovery, and reduce the incidence of complications [5–8]. A minimally invasive anterior internal pelvic fixator (INFIX) or anterior subcutaneous pelvic internal fixator (ASPIF) has been reported in the literature and involves subcutaneous placement of titanium rods and pedicle screws on the acetabulum to ensure structural stability of the anterior ring [9]. The INFIX technique is superior to anterior pelvic external fixation (EXFIX) in terms of mobility, comfort, and biomechanical stability in treatment of unstable pelvic fractures with anterior injury and eliminates the complications of needle-tract infection [10]. The anterior pedicle screw technique has been adopted, either as temporary fixation for vertically unstable pelvic fractures or as final treatment in combination with different posterior approaches.

There are many radical treatments for posterior pelvic ring instability, including percutaneous sacroiliac joint (SIJ) screws, anterior SIJ plates, posterior tension band plates, transiliac bolts, adjustable minimally invasive plates, spinal pelvic fixation (SPF) and triangular osteosynthesis (TOS) techniques. The transiliac internal fixator (TIFI) is a novel technique for stabilizing the posterior pelvic via placement of pedicle screws in the two posterior superior iliac spines in combination with a transverse rod that traverses the posterior midline of the sacrum [11]. Clinical and biomechanical studies of TIFI in the treatment of posterior pelvic injuries have been reported, and treatment indications include all types of unilateral sacral fractures and SIJ dislocations [12]. Biomechanically, some have found that TIFI is not inferior to sacroiliac screws and plates as treatment for unilateral sacral fractures of the posterior pelvic ring [13, 14]. However, TIFI is inferior to other fixation methods in biomechanical testing when using only one model of SIJ dislocation [15]. Further biomechanical and clinical studies are needed, as the test methods and models and materials used vary and are not standardized.

In one biomechanical study of the anterior INFIX technique, better stability occurred with a greater number of screws [16]. Nonetheless, this study did not compare the biomechanical

differences between the two fixation methods of placing screws on the injured and healthy sides of the pubic region.At the same time, the ligaments of the injured side of the pelvis were not removed in this study, and the posterior pelvic ring were fixed with SI screws. In addition, we have not found a biomechanical study of the anterior INFIX technique combined with the posterior TIFI technique in the treatment of Tile C pelvic fractures, nor did we retrieve any biomechanical reports on whether the diameter of the pedicle screw affects the fixation effect of unstable pelvic rings.These unknowns still require our further in-depth research and exploration.

In this study, the biomechanical stability of pedicle screws with different screw placement methods and diameters in the treatment of unstable Tile C1 pelvic fractures was evaluated and analysed by the finite element (FE) method and compared with reconstructive plate fixation. The reliability of the anterior INFIX technique combined with the posterior TIFI technique as treatment for unstable pelvic fractures was verified from the perspective of biomechanics. Additionally, the findings provide a reference for selection of the best screw placement method and screw diameter in clinical treatment.

## Materials and methods

### Data acquisition and three-dimensional (3D) geometric model establishment

The ethics committee of Tengzhou Central People's Hospital approved this study (2020-ethical review-06). A 35-year-old healthy male (height 178 cm, weight 70 kg) with no history of pelvic tumour, trauma or deformity was selected and signed the written consent before the study in May 2021. The pelvis was scanned using 64-row, 128-slice helical computed tomography (CT) (thickness 1 mm), and the data were saved in Digital Imaging and Communications in Medicine (DICOM) format as the raw data for FE mechanical analysis. The DICOM format data of the pelvic CT scans were imported into Mimics Medical 20.0 software (Materialise Company, Leuven, Belgium) for image segmentation and reconstruction, and the processed models were saved in a standard triangle language (STL) format. The modelled bony structures mainly included both the hip bones and sacrum. Each part of the STL format model was imported into the reverse engineering software Geomagic Wrap 2017 (Geomagic, Morrisville, NC, USA) for smoothing to prevent stress concentration in FE analysis. Surface reconstruction was carried out through surface automation, surface patch construction, grid construction, and surface fitting, and the model was saved in Step format. The offset function was used to offset the model inward by 1 mm. After the above process, the corresponding cancellous bone model was obtained and saved as a standard for exchange of product model data (step). The Step format model was imported into the large-scale 3D solidification software SolidWorks 2018 (SolidWorks Corp., Concord, MA, USA) for solidification; each part of the cortical bone model was established through Boolean operations, as was the corresponding cartilage model. Then, the parts were assembled to obtain a complete pelvic bone model.3D geometric models of steel plates, screws and long rods were establish in Solidworks 2018 software.

### FE model establishment and material property setting

After saving the geometric model established in SolidWorks 2018 in x_t format, it was imported into the FE analysis software Ansys Workbench 16.0 (ANSYS, Inc., Canonsburg, PA, USA). The material properties of each part of the pelvis and internal fixation materials were set, and the material was regarded as homogeneous and isotropic. The steel plates, screws, pedicle screws and long rods were made of titanium steel; the material property parameters

**Table 1. Material properties of each part of the skeleton model [17–19].**

| Materials | Young's modulus(MPa) | Poisson's ratio |
|---|---|---|
| Cortical bone (sacrum) | 6140 | 0.30 |
| Cancellous bone (sacrum) | 1400 | 0.30 |
| Cortical bone (iliac) | 17000 | 0.30 |
| Cancellous bone (iliac) | 132 | 0.20 |
| Cartilage | 54 | 0.30 |
| Titanium steel | 114000 | 0.33 |

are shown in Table 1. The ligament was approximated as a nonlinear spring unit that can only withstand tension in Ansys software. According to the anatomical starting and ending positions of the pelvic ligament, the corresponding area was selected on the model surface to create a spring unit, and the main ligament structure was reconstructed, including the anterior sacroiliac ligament (ASL), interosseous sacroiliac ligament (ISL), long posterior sacroiliac ligament (LPSL), short posterior sacroiliac ligament (SPSL), sacrospinous ligament (SSL), sacrotuberous ligament (STL), superior pubic ligament (SPL) and arcuate pubic ligament (APL). The stiffness coefficient of the main ligaments is shown in Table 2. A normal pelvis model (Fig 1A and 1B) containing ligaments was established in Ansys software and defined as Model 1. The software was used to automatically divide the model, with a control grid size of 5 mm. After grid division, the number of elements of the model was 284906, and the number of nodes was 148633. The left superior and inferior pubic ramus in the 3D geometric model of the normal pelvis was interrupted by the Segment command in the Solidworks 2018 software to simulate fracture. The left ASL, ISL, LPSL, SPSL, SSL and STL were removed to simulate left SIJ dislocation in Ansys software. A model of Tile C1 pelvic fracture (Fig 1C and 1D) was generated, which was defined as Model 2.

The 3D geometric models of steel plates, screws and long rods were moved to appropriate positions through the move and rotate commands in Solidworks 2018 software, and then Boolean operations were used to remove the overlapping parts of the bones and implants to obtain pelvic fracture fixation models.Various internal fixation repair models only retained the right hemipelvic ligaments, SPL and APL, simulating the immediate postoperative state in Ansys software. In Model 3, two 4-hole reconstruction plates were used to fix the front of the left SIJ, and an 8-hole reconstruction plate was used to fix the left superior pubic ramus fractures (Fig 2). In Models 4–15, two pedicle screws were inserted in the direction from the bilateral posterior superior iliac spine (PSIS) to the anterior inferior iliac spine (AIIS), and two pedicle screws were inserted in the direction from the bilateral AIIS to the PSIS. According to the different methods of adding screws to the pubic region, anterior pelvic ring fixation was divided as

**Table 2. Main ligament parameters of the pelvis [20, 21].**

| Ligaments | Stiffness coefficient K (N/mm) |
|---|---|
| Anterior sacroiliac ligament | 700 |
| Interosseous sacroiliac ligament | 2800 |
| Long posterior sacroiliac ligament | 1000 |
| Short posterior sacroiliac ligament | 400 |
| Sacrospinous ligament | 1400 |
| Sacrotuberous ligament | 1500 |
| Superior pubic ligament | 500 |
| Arcuate pubic ligament | 500 |

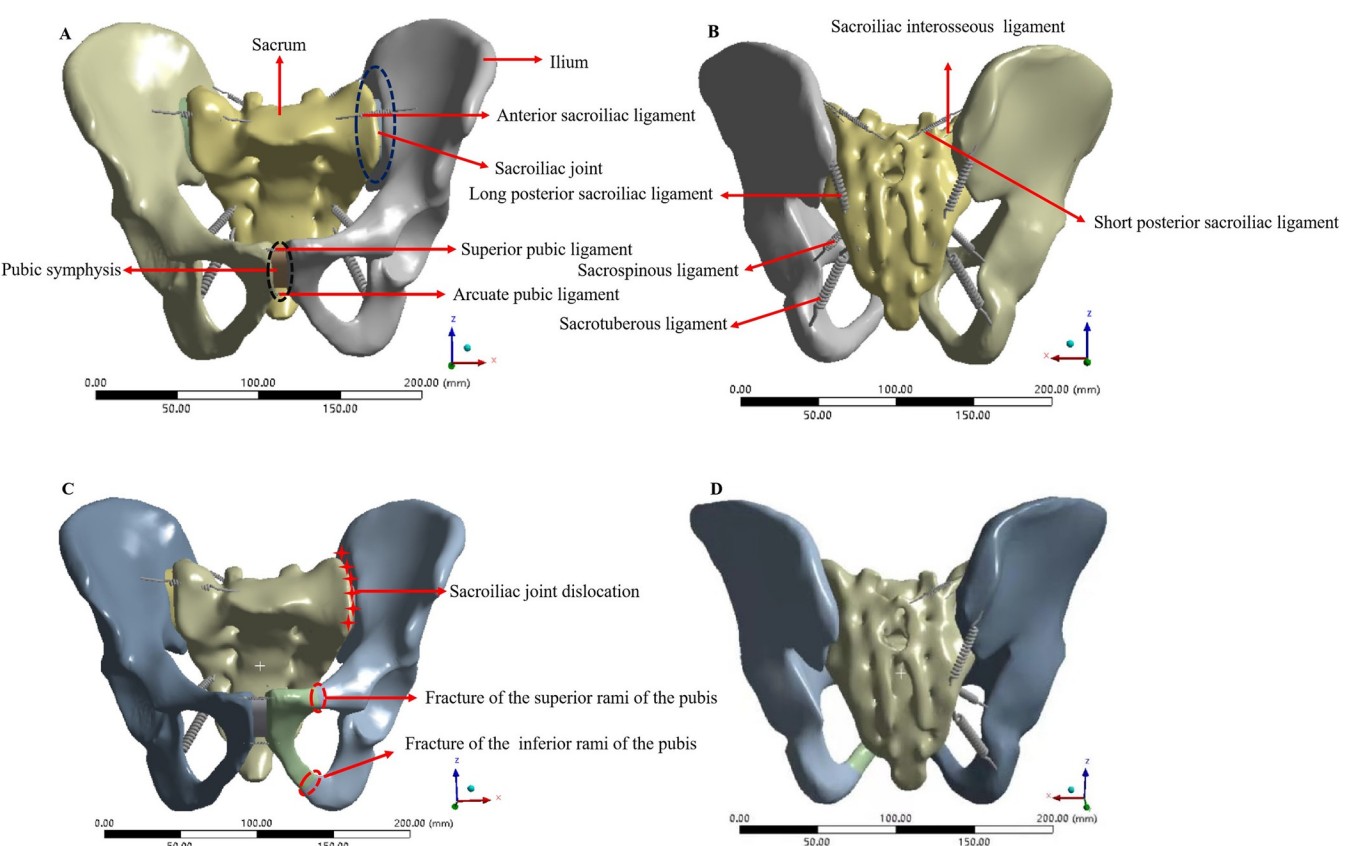

**Fig 1. Construction of the FE models of normal pelvis and A Tile C1 fractured pelvis.** (A) Anterior view of the FE model of the normal pelvis. (B) Posterior view of the FE model of the normal pelvis. (C) Anterior view of the FE model of the Tile C1 fractured pelvis. (D) Posterior view of the FE model of the Tile C1 fractured pelvis.

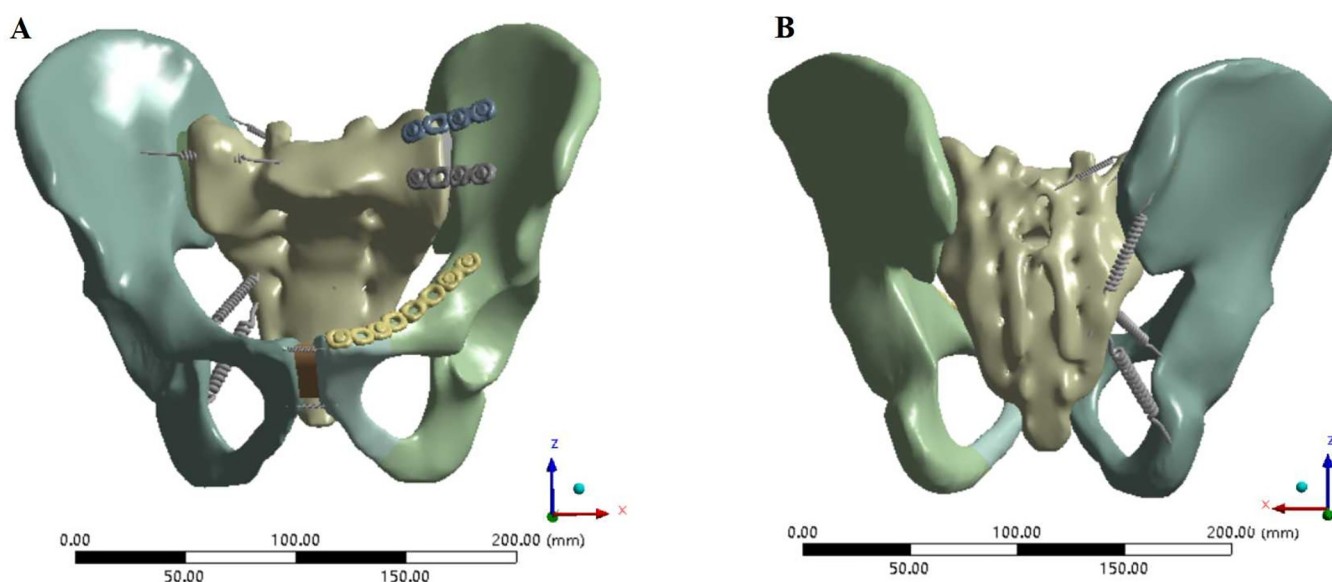

**Fig 2. FE model of anterior reconstruction plate fixation for the treatment of Tile C1 pelvic fractures.** (A) Anterior view of the FE model fixed by the reconstruction steel plate. (B) Posterior view of the FE model fixed by the reconstruction steel plate.

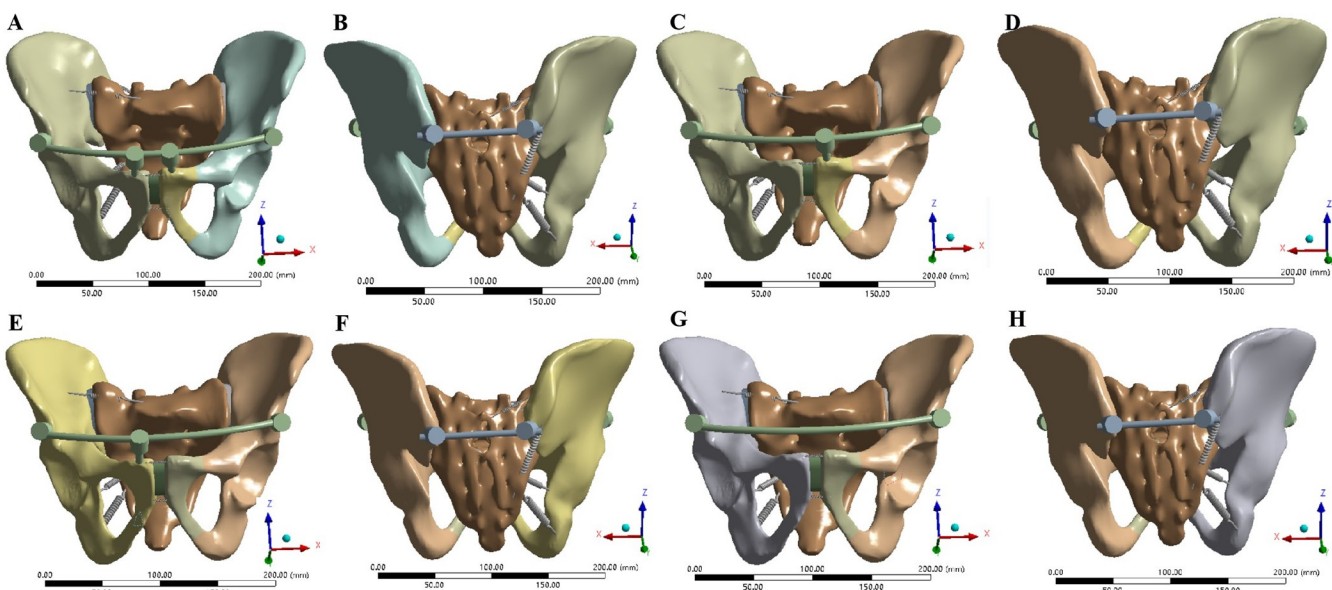

**Fig 3. FE model Tile C1 pelvic fractures fixed with pedicle screws and rods.** (A-B) Two screws in the pubic area. (C-D) One screw in the pubic area on the injured side. (E-F) One screw in the pubic area on the uninjured side. (G-H) No screws were placed in the pubic area on either side.

follows: 2 screws were added to the pubic region (method 1), 1 screw was added to the injury side of the pubic region (method 2), 1 screw was added to the healthy side of the pubic region (method 3), and no screw was added to the pubic region (method 4). The four pedicle screw placement methods are shown in Fig 3. Screws with diameters of 6.5 mm (D1), 7.0 mm (D2) and 8.0 mm (D3) were used for each screw placement method. The steel plate fixation model and the pedicle screw fixation model were automatically meshed using Ansys Workbench 16.0 software. The mesh size of the pelvis model was set to 10 mm, the mesh size of the steel plate was 1 mm, and the mesh size of the rod screw was 5 mm. After meshing, the steel plate fixed model had 446217 elements and 255952 nodes. The FE model of pedicle screw-rod fixation was grouped according to the method of screw placement in the pubic region and the diameter of the pedicle screw. The number of elements and nodes of the pedicle screw-rod fixation model are shown in Table 3.

**Table 3. FE Model grouping and number of elements and nodes for pedicle screw fixation.**

| Model | Pubic area screw placement method | Screw diameter (mm) | Number of units | Number of nodes |
|---|---|---|---|---|
| 4 | 1 | 6.5(D1) | 219946 | 115267 |
| 5 | 2 | 6.5(D1) | 219252 | 114950 |
| 6 | 3 | 6.5(D1) | 218561 | 114570 |
| 7 | 4 | 6.5(D1) | 217725 | 114169 |
| 8 | 1 | 7.0(D2) | 219526 | 115015 |
| 9 | 2 | 7.0(D2) | 218175 | 114312 |
| 10 | 3 | 7.0(D2) | 218528 | 114542 |
| 11 | 4 | 7.0(D2) | 225297 | 117837 |
| 12 | 1 | 8.0(D3) | 221358 | 115961 |
| 13 | 2 | 8.0(D3) | 220068 | 115330 |
| 14 | 3 | 8.0(D3) | 219878 | 115175 |
| 15 | 4 | 8.0(D3) | 218076 | 114204 |

## FE model load and boundary condition setting

Loads and boundary conditions were set in Ansys Workbench 16.0 software.The pubic ramus fracture interface is defined as a friction constraint with a friction coefficient of 0.3 [22], and other contact interfaces are defined as binding constraints to limit the tangential and normal displacements of the contact surfaces [16, 21–23]. The six degrees of freedom of the acetabulum were fixed to simulate the standing state of the legs. A vertical downwards load of 500 N was applied on the upper surface of the sacrum to simulate the weight of the upper body, simulate the force in a standing state, and observe the von Mises stress and displacement distribution of the pelvis.

# Results

## FE model validation of normal pelvis

In the state of standing with two legs, the normal pelvic displacement distribution was symmetrical to the left and right, with the median sacral ridge as the centre, and conducts outward in a gradually weakening wave shape. The maximal displacement of the ilium occurred at the posterior superior of the ilium wing, decreasing gradually in a wavy line towards the pubic symphysis, consistent with the findings in the literature [20]. Under a vertical downwards load of 500 N, the maximum displacement occurred at the median sacral ridge, which was $182.32 \times 10^{-3}$ mm (less than 3 mm) (Fig 4A), also similar to experimental data reported in the literature [24].

While standing on two legs, the normal pelvic stress distribution was basically symmetrical on both sides, and the overall maximum stress was 14.8 MPa, occurring near the greater sciatic notch. The stress was mainly concentrated between the greater sciatic notch and the arcuate line, anterior and superior, and posterior and superior of the acetabulum. One of the stresses was transmitted along the greater sciatic notch to the ischial rami (posterior pelvic arch); the other was transmitted along the arcuate line to the pubic ramus (anterior pelvic arch) (Fig 4B). The maximum stress value of this study was similar to the results reported by Lee et al. [25] whereby the equivalent stress value of pelvic cortical bone was 13.5~25.7 MPa under 500 N vertical load.

## Displacement analysis of the fracture fixation model

The displacement distributions of the various fixation models and the normal pelvis were basically similar but not identical. The maximum overall displacement of various fixation models was at the very end of the sacrum, the maximum displacement of the ilium appeared at the posterior and upper part of the iliac wing on the injured side, and the maximum displacement of the fracture occurred at the fracture of the inferior pubic ramus. The overall maximum displacement of the plate-fixed model (Model 3) was $601.57 \times 10^{-3}$ mm, 3.30 times that of the normal pelvis. The maximum displacement of the ilium was $319.43 \times 10^{-3}$ mm, the maximum displacement of the pubic ramus fracture was $58 \times 10^{-3}$ mm, and the maximum displacement of the internal fixation occurred at the junction of the sacral screw and the plate in the posterior pelvic ring, which was $578.44 \times 10^{-3}$ mm (Fig 5).

Among the pedicle screw rod system fixation models (Models 4–15), Model 7 had the largest overall displacement ($779.48 \times 10^{-3}$ mm), which was 4.28 times that of the normal pelvis. Conversely, Model 12 had the smallest overall maximum displacement ($691.06 \times 10^{-3}$ mm), which was 3.79 times that of the normal pelvis. The overall maximum displacement values of Models 4 to 15 were larger than those of the steel plate fixed model, and the maximum difference was less than 0.2 mm. The maximum displacement of the ilium in all pedicle screw rod system fixation models was smaller than that of the plate fixation model. In the pedicle screw

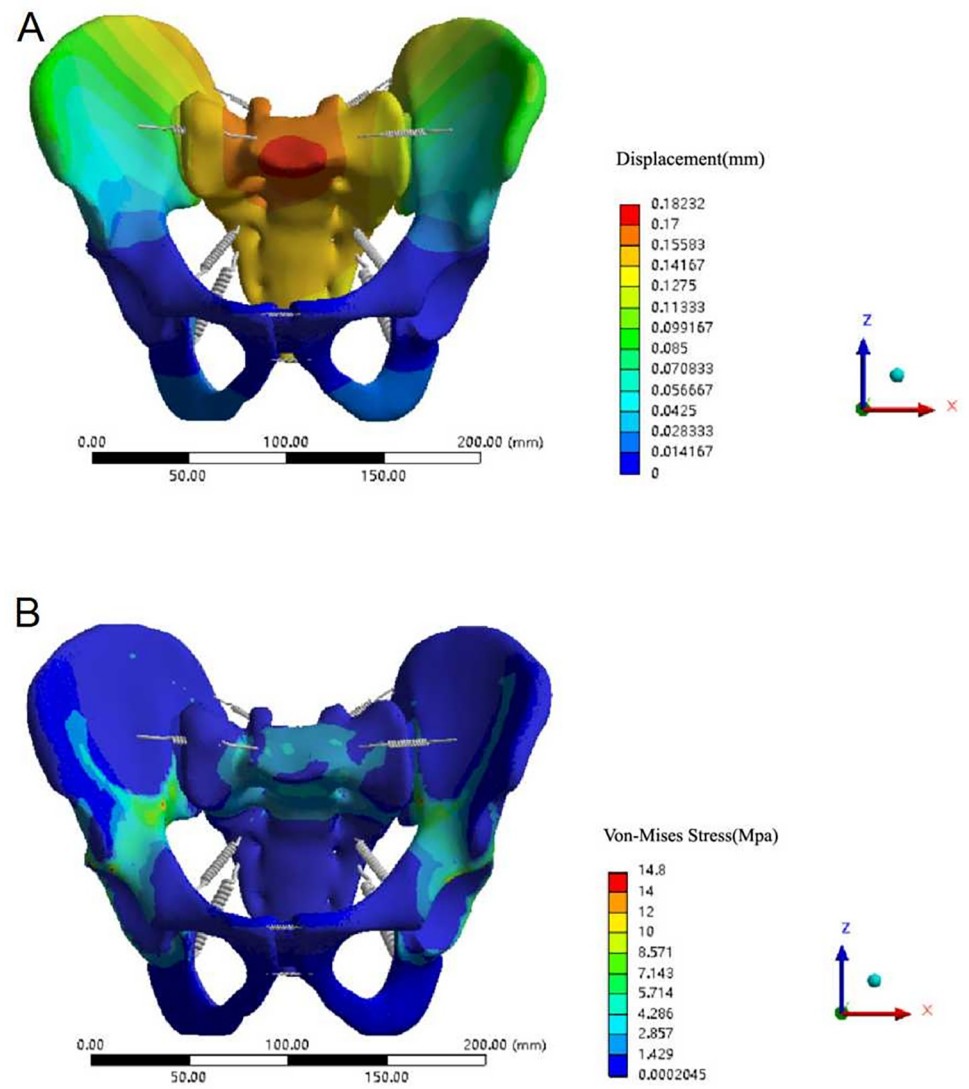

**Fig 4. Displacement and Von Mises stress distribution of the normal pelvis model.** (A) Displacement distribution of the normal pelvis model under a vertical load. (B) Von Mises stress distribution of the normal pelvis model under a vertical load.

rod system fixation model, the maximum displacement of the fracture of the pubic ramus was smaller than that of the plate fixation model, except for Model 7, but the difference was less than 0.01 mm. The maximum displacement of internal fixation with all pedicle screw rod system fixation models occurred at the pedicle screw-rod junction on the injured side of the posterior pelvis, and the displacement distribution of the posterior fixation rod gradually decreased from the injured side to the healthy side and from the end cap to the tip. The maximum displacement of the internal fixation was smaller than that of the steel plate fixation model. In the case of the same anterior pelvic ring screw placement, the maximum displacement of the whole pelvis, iliac crest, pubic ramus fracture and internal fixation was as follows: D1 > D2 > D3. In the case of the same pedicle screw diameter, the maximum displacement of the pelvis as a whole, iliac crest, pubic ramus fracture and internal fixation was ranked as follows: method 1 < method 2 < method 3 < method 4 (Figs 6 and 7).

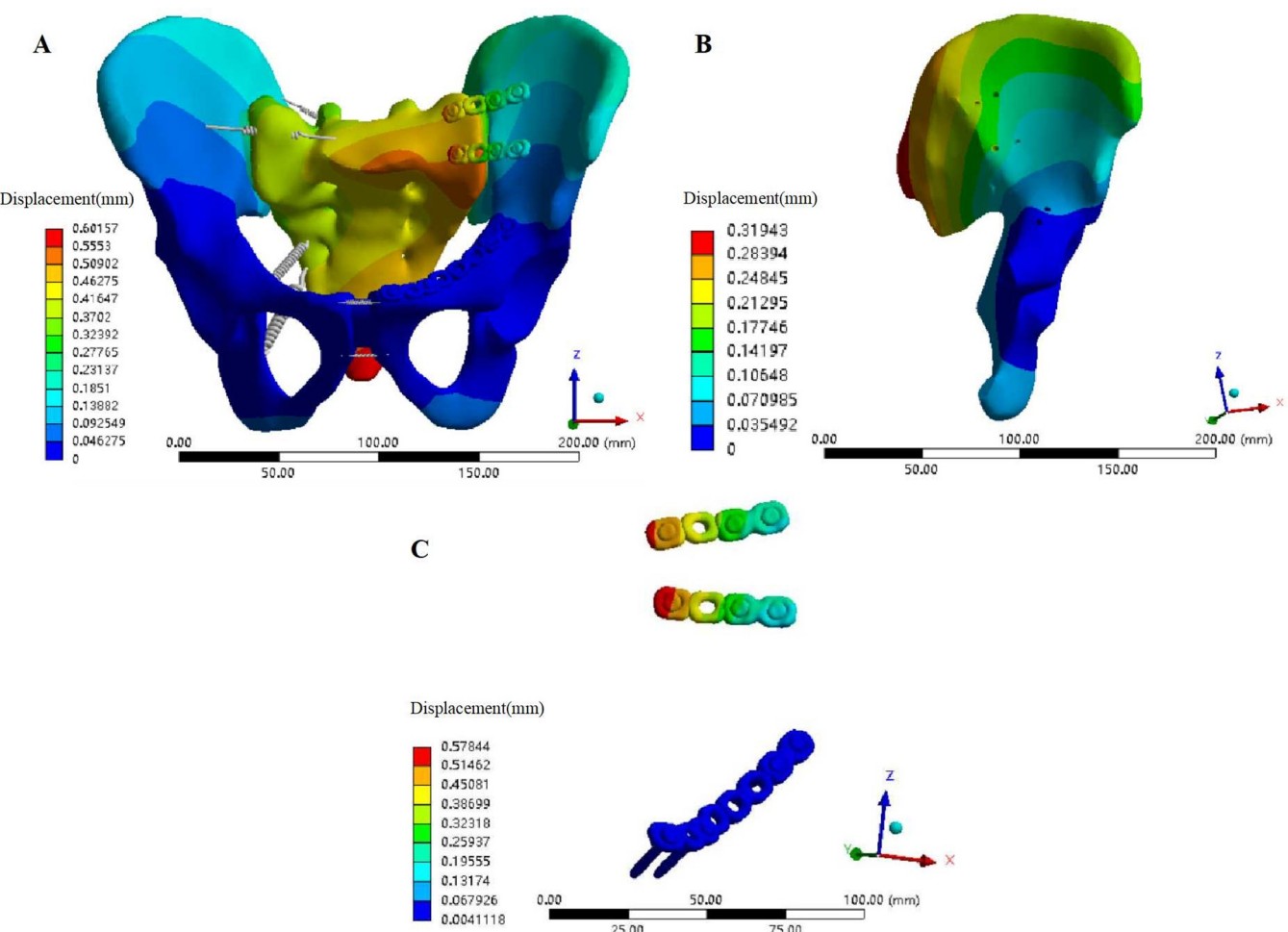

**Fig 5. Displacement distribution of each component of the pelvis model fixed with steel plate.** (A) Displacement distribution of the pelvis model fixed with steel plate under a vertical load. (B) Displacement distribution of the iliac bone of the pelvic model fixed with a steel plate under a vertical load. (C) Displacement distribution of the steel plate and screw of the pelvic model fixed by a steel plate under a vertical load.

### Stress analysis of the fracture fixation model

Although the stress distribution of the various fixation models and normal pelvis was similar, the stress distribution between each fixation model and the normal pelvis model was not exactly the same. For the plate fixation model (Model 3), the overall maximum stress appeared at the junction of the sacral screw and the plate in the posterior ring of the pelvis; the value was 455.1 MPa, which was 30.75 times that of the normal pelvis (Fig 8A). The maximum stress of the ilium was 19.01 MPa, appearing near the greater sciatic notch on the injured side. The maximum stress at the fracture of the pubic ramus was 0.9 Mpa (Fig 8B).

The overall maximum stress of the pedicle screw rod fixation model (Models 4–15) was also the maximum stress of the ilium, and both occurred near the greater sciatic notch on the injured side. The overall maximum stress of Model 7 was the largest (14.6 MPa) the overall maximum stress of Model 12 the smallest (12.7 MPa); the overall maximum stress of Models 4–15 was smaller than that of the normal pelvis model and the steel plate fixation model. The maximum stress of internal fixation occurred at the junction of the pedicle screw and the ilium on the injured side of the posterior pelvic ring. Model 7 showed the largest internal

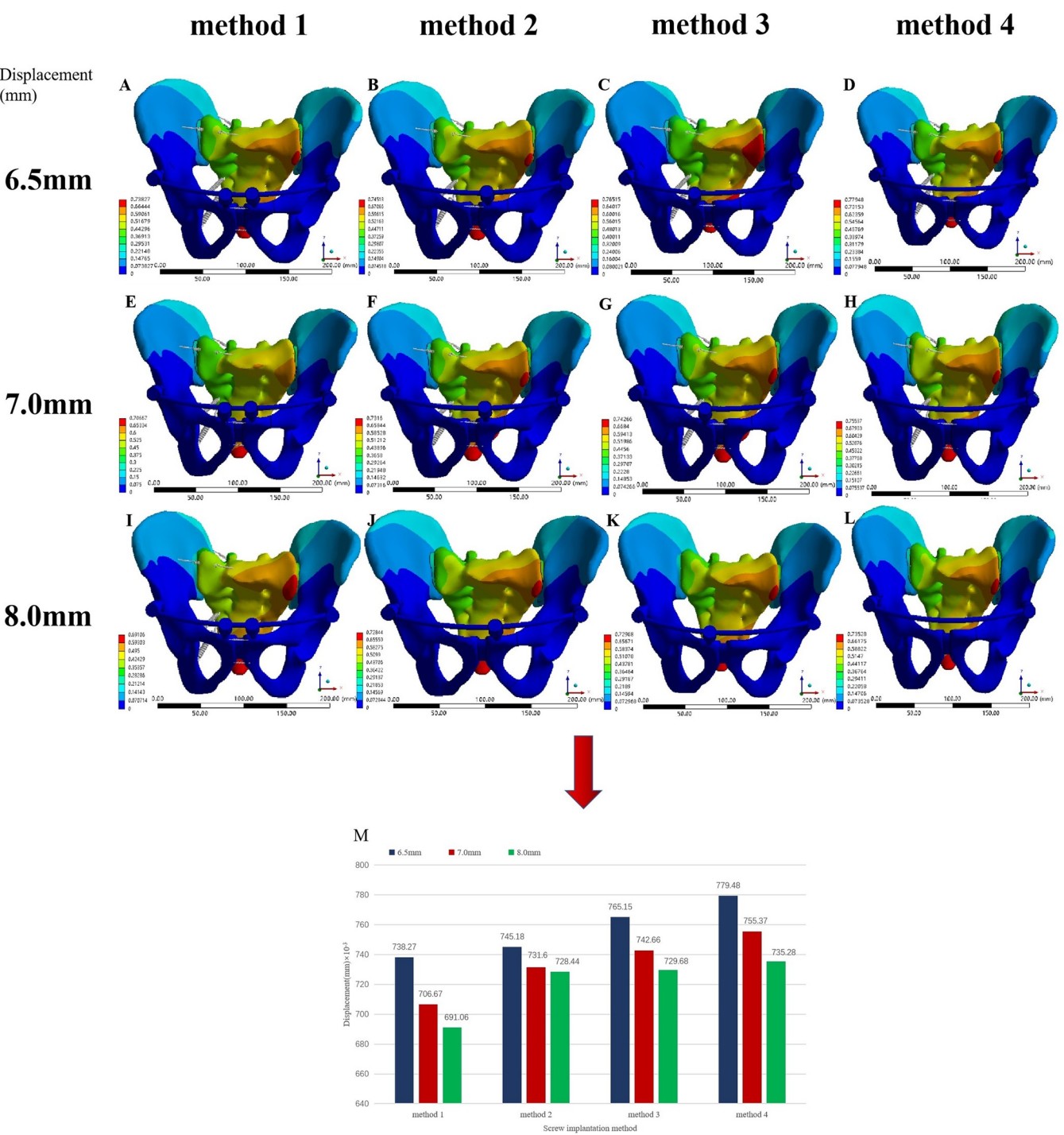

**Fig 6. Displacement distribution of pelvic models fixed with pedicle screws under a vertical load.** (A) Model 4. (B) Model 5. (C) Model 6. (D) Model 7. (E) Model 8. (F) Model 9. (G) Model 10. (H) Model 11. (I) Model 12. (J) Model 13. (K) Model 14. (L) Model 15, (M) Comparison of the maximum displacement of the pelvic model with pedicle screws of three diameters and four fixation methods under a vertical load.

fixation stress (13.23 MPa) and Model 12 the smallest overall maximum stress (11.7 MPa); its value was much smaller than that of the steel plate fixation model. In the case of the same anterior pelvic ring screw placement, the maximum stress of the whole pelvis (ilium) and internal

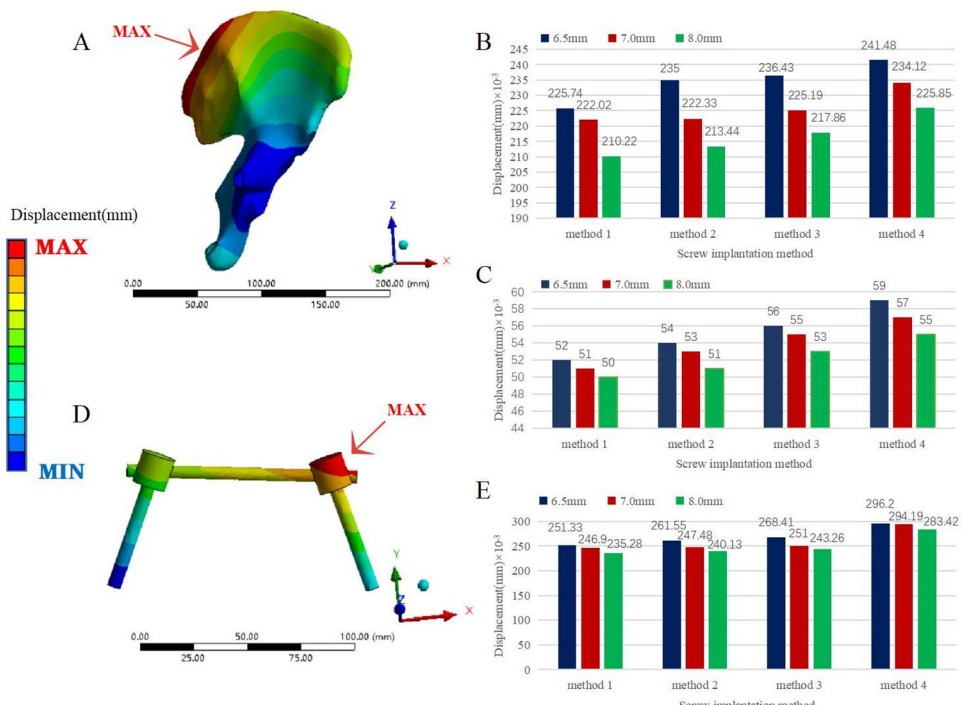

**Fig 7. Displacement comparison of pedicle screw fixation models with different diameters and fixation methods.**
(A) Displacement distribution of the iliac bone of the pelvis model fixed with pedicle screws under a vertical load. (B) Comparison of the maximum displacement of the iliac bone of the pelvic model with pedicle screws of three diameters and four fixation methods under a vertical load. (C) Comparison of the maximum displacement of the pubic ramus fracture of the pelvic model with pedicle screws of three diameters and four fixation methods under a vertical load. (D) Displacement distribution of the pedicle screw rod in the posterior ring of the pelvic model fixed with pedicle screws under a vertical load. (E) Comparison of the maximum displacement of the pedicle screw rod in the posterior ring of the pelvic model with pedicle screws of three diameters and four fixation methods under a vertical load.

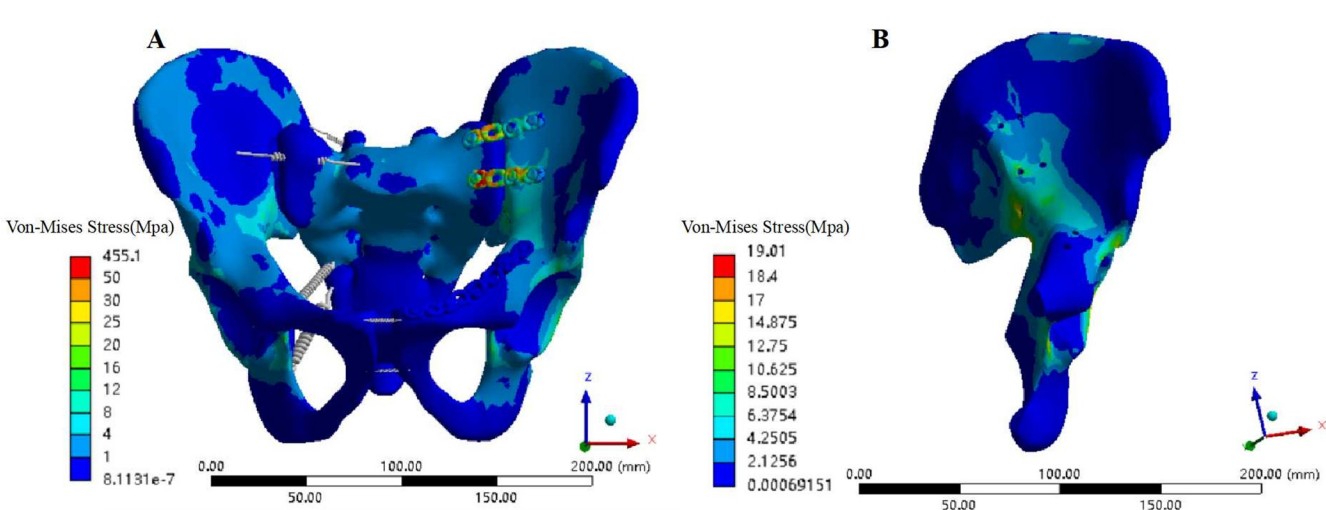

**Fig 8. Von Mises stress distribution of each component of the pelvis model fixed with steel plate.** (A) Von Mises stress distribution of the pelvis model fixed with a steel plate under a vertical load. (B) Von Mises stress distribution of the iliac bone of the pelvic model fixed with a steel plate under a vertical load.

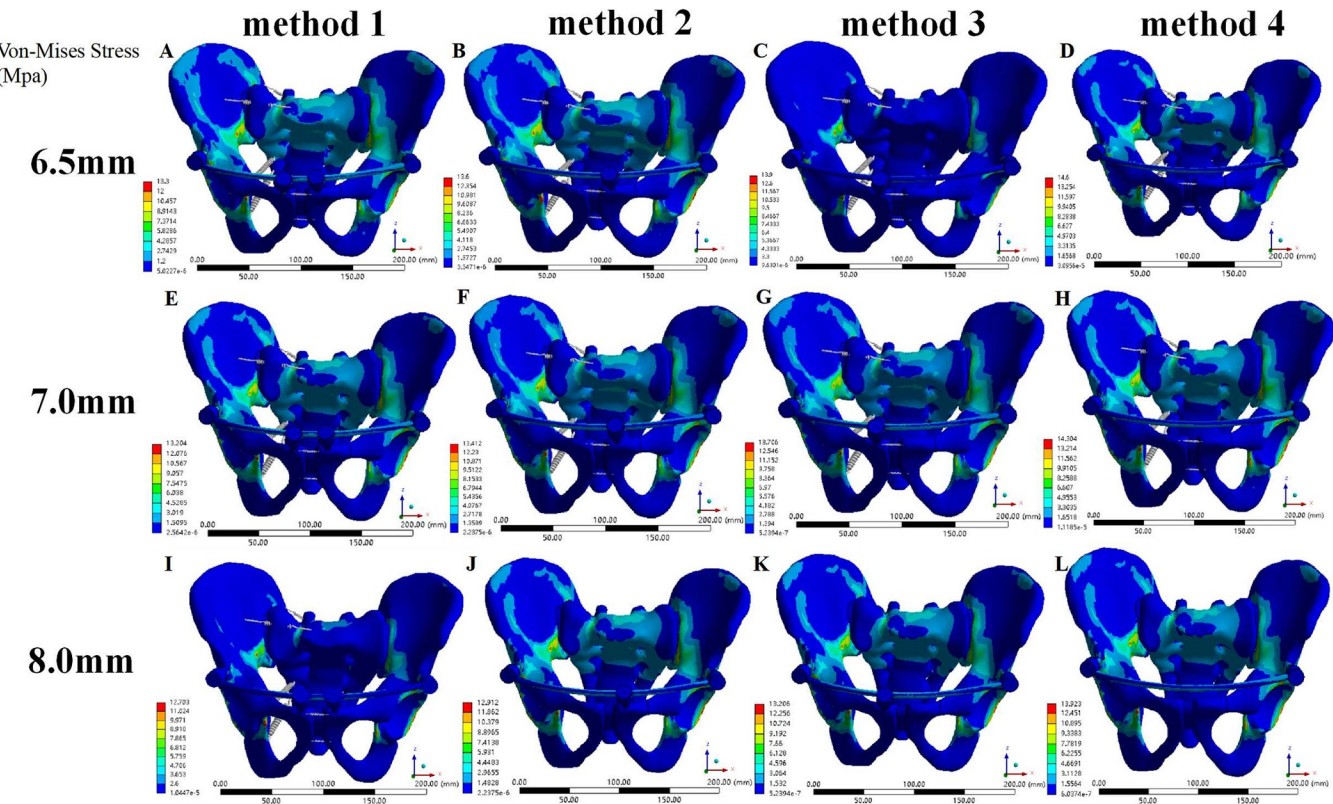

**Fig 9. Von Mises stress distribution of pelvic models fixed with pedicle screws under a vertical load.** (A) Model 4. (B) Model 5. (C) Model 6. (D) Model 7. (E) Model 8. (F) Model 9. (G) Model 10. (H) Model 11. (I) Model 12. (J) Model 13. (K) Model 14. (L) Model 15.

fixation was as follows: D1 > D2 > D3. In the case of pedicle screw placement with the same diameter, the maximum stress of the whole pelvis (ilium) and internal fixation was ranked as follows: method 1 < method 2 < method 3 < method 4. The maximum stress difference at the fracture of the pubic ramus between the pedicle screw-rod fixation models and between the pedicle screw-rod model and the plate fixation model was less than 1 MPa (Figs 9 and 10).

## Discussion

The ideal treatment for vertically rotationally unstable pelvic fractures remains controversial because determining the best treatment approach is a multifactorial process. As the innovation and minimally invasive concept of internal fixation devices has gradually gained attention, the technology that can reduce soft tissue trauma, operation time, and iatrogenic nerve and blood vessel damage caused by surgical exposure while providing sufficient biomechanical stability has also been a topic of discussion among traumatologists.SI screw fixation is currently a popular posterior fixation method for the treatment of unstable pelvic fractures, but this fixation method has the disadvantages of high technical requirements, repeated fluoroscopy, and high rate of vascular and nerve injury. Anterior SIJ plate fixation is one of the standard methods for the treatment of unstable posterior pelvic ring injuries, but it requires open reduction, and extensive soft tissue dissection can easily lead to complications such as massive pelvic bleeding and infection. Posterior tension band plate fixation reduces intraoperative fluoroscopy time and the risk of nerve or vascular injury, and can achieve functional results similar to SI screw fixation. However, it is sometimes difficult to achieve the correct bending of the plate during

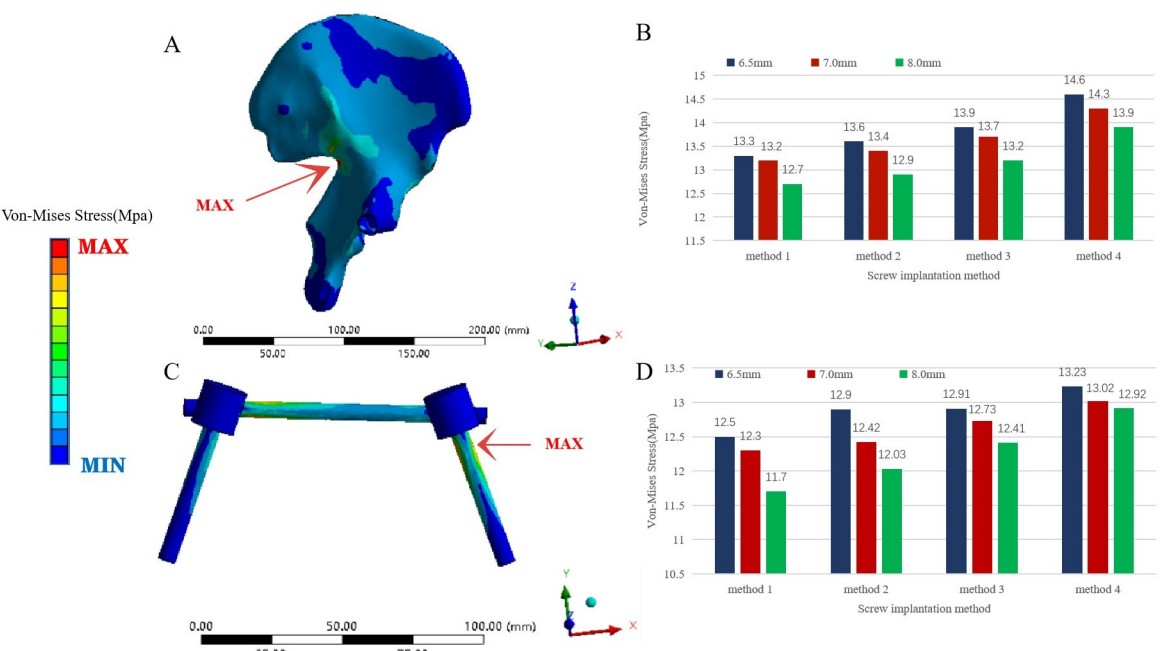

**Fig 10. Von Mises stress comparison of pedicle screw fixation models with different diameters and fixation methods.** (A) Von-Mises Stress distribution of the iliac bone of the pelvis model fixed with pedicle screws under a vertical load. (B) Comparison of the maximum Von-Mises Stress of the iliac bone of the pelvic model with pedicle screws of three diameters and four fixation methods under a vertical load. (C) Von-Mises Stress distribution of the pedicle screw rod in the posterior ring of the pelvic model fixed with pedicle screws under a vertical load. (D) Comparison of the maximum Von-Mises Stress of the pedicle screw rod in the posterior ring of the pelvic model with pedicle screws of three diameters and four fixation methods under a vertical load.

the operation. Repeated bending of the plate may reduce its strength and even damage the threads of the threaded holes.

Drawing on the principle of posterior tension band plate fixation of the SIJ, Füchtmeier et al. [26] analysed 27 patients with unstable C-type pelvic fractures (AO classification) (5 cases of unilateral SIJ dislocation and 22 cases of unilateral sacral fractures) treated with TIFI and found hat TIFI provides stability comparable to the use of two sacroiliac screws, with higher stiffness and lower stress but without the risk of excessive compression even with sacral transforaminal comminuted fractures. Wu et al. [27] and Hua et al. [28] used INFIX for the anterior ring and TIFI for the treatment of unstable Tile C pelvic ring fractures in clinical studies and reported that iliac screws and plates have the same clinical effect. TIFI internal fixation in the treatment of unstable posterior pelvic rings has the advantages of being a minimally invasive, simple operation with reduced blood loss and with a low risk of iatrogenic vascular and nerve injury. The main indication for the current TIFI technique is all unilateral instability of the posterior pelvic ring, with the ideal indication being a unilateral SIJ rupture or a unilateral sacral fracture with mild or moderate dislocation [26].

Although TIFI has sufficient mechanical stability for the treatment of sacral fractures, there is still controversy regarding the biomechanics of isolated TIFI in the treatment of SIJ destruction. Vigdorchik et al. [15] used synthetic bone to model SIJ destruction and found that a single TIFI fixation without any additional screws was less stable. Ueno et al. [29] performed a biomechanical study using polyethylene model material to simulate unilateral SIJ dislocation, observing that one set of TIFI devices showed poor mechanical results but that two sets had better mechanical stability. Using a synthetic bone model, Shinohara et al. [30] found that posterior fixation using a modified dual TIFI device was biomechanically stronger and stiffer than

conventional posterior plate fixation. Using a freshly frozen human pelvis to simulate an AO C-type injury model (symphysis pubis separation and unilateral SIJ destruction) to compare TIFI, two sacroiliac screws, and two V-positioned ventral plates, Dienstknecht et al. [13] showed that the stability of the fixed SIJ had the same biomechanical stability of the three internal fixation methods in the single-leg standing state, and it was considered that TIFI may be a suitable substitute for other implants. The authors also suggested that whether the implants were sufficiently anchored in the model had not been determined in previous Sawbone models. In this study, a normal pelvis model and a Tile C1 pelvic fracture model (unilateral SIJ dislocation and superior and inferior pubic ramus fractures) were established by the FE method, and FE modelling was verified by comparing the normal pelvis model with previous FE studies. We fixed the pelvic fracture models with two internal fixators. All reduction and fixation models were considered anatomical reduction, and all models in our study had a maximum displacement of less than 1 mm. According to previous studies [31–33], any displacement greater than 10 mm is a poor prognostic indicator, motion displacement of 0.1 to 1 mm can improve healing time, and displacement less than 5 mm can have a higher functional score; the smaller the maximum displacement is, the higher the stability is. Therefore, we believe that fixing the anterior and posterior pelvic rings with a pedicle screw rod system achieves sufficient mechanical stability. The overall stability of the plate fixation model was slightly higher than that of the pedicle screw fixation model, and the stability of anterior pelvic ring fractures was similar. However, anterior SIJ plate fixation causes the stress of the entire pelvis to be excessively concentrated at the junction of the plate and screw on the sacral side, which also increases the risk of plate and screw failure to a certain extent. The stability of the ilium and internal fixation in the pedicle screw system fixation model was higher than that of the plate fixation model, which we believe was due to the large diameter of the pedicle screw fixed in the hard iliac compact bone area and the long pedicle rod, as well as the dispersion of stress caused by long screws. According to the study of Liu et al. [16], the biomechanical properties of the fixation model gradually improve with an increase in fixation screws in the anterior ring stent, which is consistent with our findings. In addition, we found that for the pedicle screw rod system fixation model, the larger the screw diameter was, the greater the stability of the pelvis was. Vaidya et al. [34] reported loss of reduction in 3 of 91 patients with pelvic fractures treated with 7–8.5 mm diameter pedicle screws and lateral femoral cutaneous nerve palsy in 27 patients (30%). However, Scheyerer et al. [35] reported no development of lateral femoral cutaneous nerve palsy using pedicle screws with a diameter of 6.5 mm. Although Hoskins et al. [36] found no loss of reduction when pedicle screws with a diameter of 10 mm were used in the treatment of pelvic fractures, the incidence of lateral femoral cutaneous nerve palsy was 48%, which was larger than the diameter of the screws used in our study. Large-diameter screws increase fixation stability but also the risk of irritation of the lateral femoral cutaneous nerve. In this study, the stress on the internal fixator of the pedicle screw rod fixation system was mainly concentrated on the end cap of the screw on the injured side of the posterior pelvic ring. Failure to tighten the end cap of the screw during the operation will easily lead to its loosening, causing implantation. Risk of entry failure was also validated in the clinical follow-up by Steer et al. [37].

The main ligaments of the pelvic structure, including the interosseous ligaments, the short posterior sacroiliac ligaments, the sacrotuberous ligaments, and the sacrospinous ligaments, are considered in FE computer simulations, and these ligaments are tightened during nutation of the sacrum to attach the posterior ilium pulled together, which increases compression of the SIJ; the force is closed, improving the clinical and biomechanical outcomes of SIJ surgery. Tile C pelvic injuries are both vertically and rotationally unstable. In addition to posterior fixation, our fixation for pubic ramus fractures reduced the stress level of the pelvic and fixation

systems, changing the pelvic instability to a certain extent. The bilateral iliac screws travel in the bony channel between the inner and outer plates of the ilium from the posterior superior iliac spine to the anterior inferior iliac spine, creating a large torque against vertical displacement and sagittal rotation. The SIJ surface has a rough texture, complementary ridges and grooves, and a range of friction coefficients on the articular surface. Anatomical reduction of the SIJ allows for perfect sacrum embedding in the pelvis, whereas the anterior INFIX technique allows for indirect compression of the SIJ, and the lateral and frictional forces of joint compression are necessary for the joint to withstand vertical loads [38]. Considering that the friction coefficient is also an important factor affecting the load transfer in the SIJ contact model, and the smooth joint surface model combined with the lower friction coefficient may result in a large deviation from the actual model.We adopted the binding constraint of the SIJ to simplify the interlocking structure and friction coefficient of the SIJ. The setting also increased the stability of the SIJ in our model against shear force to a certain extent and reduced stress in the model.

The limitations of this study are as follows. 1. Due to the difficulty in finding matching and reliable data on muscle strength [30], the effect of muscle strength and synovial conditions on pelvic stability was not considered in our research model. 2. Muscles such as the rectus femoris, sartorius, iliacus, gluteus maximus, and hamstrings are thought to be involved in SIJ force closure, and these muscles influence SIJ movement through appropriate lever arms. Since these muscle forces were omitted, we only assessed pelvic stability when standing on two legs and not the most unstable situation when standing on one leg, as these muscles have a greater role in providing trunk and SIJ stability biomechanical advantages. 3. Our research explored a treatment method for Tile C1 pelvic fractures with a short learning curve, minimal invasiveness, few complications and sufficient mechanical stability instead of finding the most stable treatment method.

## Conclusions

According to our data, the use of anterior INFIX in combination with posterior TIFI in the treatment of Tile C pelvic fractures achieves adequate biomechanical effects similar to those of plate fixation, rendering it a suitable alternative to other implants. In the treatment of pedicle screw rod fixation, the greater the diameter and number of pedicle screws were, the greater the pelvic stability was, and the greater was the stability of the screws placed on the injured side of the pubic region than the screws on the healthy side,also providing a biomechanical reference for rational selection of pedicle screw placement methods and diameters in clinical practice.

## Acknowledgments

We would like to thank Professor Man J of School of Mechanical Engineering of Shandong University for his valuable suggestions to improve the process in the analysis and interpretation of the results.

## Author Contributions

**Conceptualization:** Yuanzheng Song, Ximing Yang, Feng Lin.

**Data curation:** Yuanzheng Song, Ximing Yang, Feng Lin.

**Formal analysis:** Yuanzheng Song, Ximing Yang.

**Funding acquisition:** Yuanzheng Song.

**Investigation:** Changsheng Shao.

**Methodology:** Yuanzheng Song, Changsheng Shao.

**Project administration:** Yuanzheng Song, Ximing Yang.

**Resources:** Yuanzheng Song.

**Software:** Yuanzheng Song.

**Supervision:** Changsheng Shao, Feng Lin.

**Validation:** Yuanzheng Song, Changsheng Shao.

**Writing – original draft:** Yuanzheng Song.

**Writing – review & editing:** Yuanzheng Song, Ximing Yang, Feng Lin.

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
