## [Decision Letter · Decision Letter 0]

27 Jun 2022

PONE-D-22-07190Biomechanical study of anterior and posterior pelvic rings using pedicle screw fixation for Tile C1 pelvic fractures: finite element analysisPLOS ONE

Dear Dr. Yang,

Thank you for submitting your manuscript to PLOS ONE. After careful consideration, we feel that it has merit but does not fully meet PLOS ONE’s publication criteria as it currently stands. Therefore, we invite you to submit a revised version of the manuscript that addresses the points raised during the review process.

We look forward to receiving your revised manuscript.

Kind regards,

Osama Farouk

Academic Editor

PLOS ONE

Journal Requirements:

6. Please amend your list of authors on the manuscript to ensure that each author is linked to an affiliation. Authors’ affiliations should reflect the institution where the work was done (if authors moved subsequently, you can also list the new affiliation stating “current affiliation:….” as necessary).

7. Your ethics statement should only appear in the Methods section of your manuscript. If your ethics statement is written in any section besides the Methods, please delete it from any other section. 

Reviewers' comments:

Reviewer's Responses to Questions

**Comments to the Author**

1. Is the manuscript technically sound, and do the data support the conclusions?

Reviewer #1: Yes

Reviewer #2: Yes

2. Has the statistical analysis been performed appropriately and rigorously? 

Reviewer #1: Yes

Reviewer #2: Yes

3. Have the authors made all data underlying the findings in their manuscript fully available?

Reviewer #1: Yes

Reviewer #2: Yes

4. Is the manuscript presented in an intelligible fashion and written in standard English?

Reviewer #1: Yes

Reviewer #2: Yes

5. Review Comments to the Author

Reviewer #1: Pelvic injuries with biomechanically unstable situations are a challenge in modern traumatology. This makes safe and early surgical treatment all the more necessary. For optimal placement and selection of required screws and plates, biomechanical studies with finite element analysis are helpful. Results of these studies show that the restoration is more resilient than patients are currently allowed in the postoperative setting. The studies allow calculation of the safety margin, which is very high with good restoration.

In this context, Yuanzheng Song's group of authors presents their biomechanical work on pedicle screws in Tile-C1 pelvic fractures. Using the finite element simulation technique, they analyze the positioning of the screws and compare them with a plate fixation model. The three-dimensional digital model was subjected to a vertical load of 500 N on the top of the sacrum on the computer to calculate the displacement and stress distribution of the pelvis in standing. As a result, the maximum displacement in the ISG was less than 1 mm.The maximum displacement difference and the maximum stress difference were less than 0.01 mm and 1 MPa, respectively. They conclude that anterior and posterior screw fixation achieves sufficient biomechanical stability and can be used as a suitable alternative to other implants. The larger the diameter and number of pedicle screws were, the greater the pelvic stability was, and the greater the stability of the screws placed on the injured side of the pubic region was than that of the screws on the healthy side.

The limitations of the study are the points listed by the authors and the fact that this is a calculated model that must first prove itself in clinical practice. In particular, this may be a challenge in older people with FFP-type pelvic injuries.

It is not clear to the reviewer how this study differs from the known studies and what the advantages of this work are. Since 2015, a number of biomechanical finite element analysis studies have been published on the treatment of C1 fractures. Some of them have an almost identical study design, figures and results.

The authors must please revise their manuscript to make it clear what the advantage of publishing their work is compared to the known publications.

Reviewer #2: This reviewing study was conducted to determine Biomechanical study of anterior and posterior pelvic rings using pedicle screw fixation for Tile C1 pelvic fractures: finite element analysis. The article appeared to be an interesting. However there are some points take in considers including the following:

1) Determine the synchronization between the software that was mentioned in the chapter of materials and methods and the models in a way that how can the input data interpret and explained by the mentioned software?

2) What are the limitation of the current study that it should be mentioned in the introduction of the present study?

3) The expected using of different type of internal fixation for the cases of fractured pelvic can de added and brief description of using them in these conditions?

6. PLOS authors have the option to publish the peer review history of their article (what does this mean?). If published, this will include your full peer review and any attached files.

Reviewer #1: No

Reviewer #2: No

---

## [Author Response · Author response to Decision Letter 0]

10 Jul 2022

Dear reviewers,

Thank you so much for your very positive feedback on our study and investing time to provide detailed and insightful comments to improve our paper. Reviewers’ comments were taken carefully and revisions were made accordingly. Edited contents were highlighted in red in the main text. Looking forward to from you soon!

Respond to reviewer#1

Dear reviewer,

Thank you so much for your very positive feedback on our study！I quite agree with you that this is a calculated model that must first prove itself in clinical practice.We have validated this technique in clinical work with good postoperative results, and this study provides biomechanical evidence for the use of pedicle screws for fixation of anterior and posterior pelvic rings.There are also some literature reports on the clinical application effect of this technology.References are as follows:

Wu XT, Liu ZQ, Fu WQ, Zhao S. Minimally invasive treatment of unstable pelvic ring injuries with modified pedicle screw-rod fixator. J Int Med Res. 2018;46:368–380. 

Hua X, Yan SG, Cui Y, Yin Z, Schreiner AJ, Schmidutz F. Minimally invasive internal fixator for unstable pelvic ring injuries with a pedicle screw-rod system: a retrospective study of 23 patients after 13.5 months. Arch Orthop Trauma Surg. 2019;139:489–496.

C.Bi, Q. G. Wang, C. Nagelli, J. H. Wu, Q. Wang and J. D. Wang.Treatment of Unstable Posterior Pelvic Ring Fracture with Pedicle Screw-Rod Fixator Versus Locking Compression Plate: A Comparative Study.Med Sci Monit.2016;22:3764-3770. 

We believe that the fixation of the implant in osteoporotic bone can be increased by cement augmentation in older people with FFP-type pelvic injuries.(Schmitz, P., Baumann, F., Acklin, Y.P. et al. Clinical application of a minimally invasive cement-augmentable Schanz screw rod system to treat pelvic ring fractures. International Orthopaedics (SICOT) 43, 697–703 (2019).https://doi.org/10.1007/s00264-018-3988-6.)

Since 2015, a number of biomechanical finite element analysis studies have been published on the treatment of C1 fractures. Nonetheless, we have not found a biomechanical study of the anterior INFIX technique combined with the posterior TIFI technique in the treatment of Tile C pelvic fractures, nor did we retrieve any biomechanical reports on whether the diameter of the pedicle screw affects the fixation effect of unstable pelvic rings.In one biomechanical study of the anterior INFIX technique, better stability occurred with a greater number of screws(Liu L, Fan S, Chen Y, Peng Y, Wen X, Zeng D, et al. Biomechanics of anterior ring internal fixation combined with sacroiliac screw fixation for tile C3 pelvic fractures. Med Sci Monit. 2020;26:e915886). Nonetheless, this study did not compare the biomechanical differences between the two fixation methods of placing screws on the injured and healthy sides of the pubic region.At the same time, the ligaments of the injured side of the pelvis were not removed in this study, and the posterior pelvic ring were fixed with SI screws.Our study supplements the gaps in these previous studies, which is where this study distinguishes the published literature and is also the innovation of this paper.Related description has been stated and added in the introduction section.(page 5-6，line 80-95).

Thanks again for your comments on our study.

Respond to reviewer #2

1.Thank you so much for your very positive feedback on our study！In order to ensure the synchronization between the software and the model, some modifications have been made in the Materials and methods section of this paper, and the modified parts have been marked with red fonts.（page 7-8,line 120-144;page 9,line 155-160;page 11,line 191）

2.Thank you so much for your very positive feedback on our study! We have added the limitations of the current study in the introduction of the present study and marked them in red.（page 5-6,line 81-88）

3.Thank you so much for your very positive feedback on our study！We have added several commonly used methods of stabilizing the posterior pelvic ring and described their use in the Discussion section of the article. Additions have been marked in red font.SI screw fixation is currently a popular posterior fixation method for the treatment of unstable pelvic fractures, but this fixation method has the disadvantages of high technical requirements, repeated fluoroscopy, and high rate of vascular and nerve injury. Anterior SIJ plate fixation is one of the standard methods for the treatment of unstable posterior pelvic ring injuries, but it requires open reduction, and extensive soft tissue dissection can easily lead to complications such as massive pelvic bleeding and infection. Posterior tension band plate fixation achieved functional results consistent with SI screw fixation with reduced intraoperative fluoroscopy time without the risk of nerve or vascular damage associated with SI screw fixation. However, it is sometimes difficult to achieve the correct bending of the plate during the operation. Repeated bending of the plate may reduce its strength and even damage the threads of the threaded holes.（page 18-19,line 327-337）

Thanks again for your comments on our study.

---

## [Decision Letter · Decision Letter 1]

8 Aug 2022

Biomechanical study of anterior and posterior pelvic rings using pedicle screw fixation for Tile C1 pelvic fractures: finite element analysis

PONE-D-22-07190R1

Dear Dr. Yang,

We’re pleased to inform you that your manuscript has been judged scientifically suitable for publication and will be formally accepted for publication once it meets all outstanding technical requirements.

Kind regards,

Osama Farouk

Academic Editor

PLOS ONE

Additional Editor Comments (optional):

Reviewers' comments:

Reviewer's Responses to Questions

**Comments to the Author**

1. If the authors have adequately addressed your comments raised in a previous round of review and you feel that this manuscript is now acceptable for publication, you may indicate that here to bypass the “Comments to the Author” section, enter your conflict of interest statement in the “Confidential to Editor” section, and submit your "Accept" recommendation.

Reviewer #1: All comments have been addressed

Reviewer #2: All comments have been addressed

2. Is the manuscript technically sound, and do the data support the conclusions?

Reviewer #1: Yes

Reviewer #2: Yes

3. Has the statistical analysis been performed appropriately and rigorously? 

Reviewer #1: Yes

Reviewer #2: Yes

4. Have the authors made all data underlying the findings in their manuscript fully available?

Reviewer #1: Yes

Reviewer #2: Yes

5. Is the manuscript presented in an intelligible fashion and written in standard English?

Reviewer #1: Yes

Reviewer #2: Yes

6. Review Comments to the Author

Reviewer #1: The authors have carefully revised the manuscript and clarified and explained the critical points. Thus, there is nothing against the publication of this interesting article.

Reviewer #2: accepted this paper to be published as this paper meet the technical sound The article appeared to be an interesting

7. PLOS authors have the option to publish the peer review history of their article (what does this mean?). If published, this will include your full peer review and any attached files.

Reviewer #1: **Yes: **Christoph Biehl

Reviewer #2: No

---

## [Editor Report · Acceptance letter]

12 Aug 2022

PONE-D-22-07190R1 

Biomechanical study of anterior and posterior pelvic rings using pedicle screw fixation for Tile C1 pelvic fractures: Finite element analysis 

Dear Dr. Yang:

I'm pleased to inform you that your manuscript has been deemed suitable for publication in PLOS ONE. Congratulations! Your manuscript is now with our production department. 

Kind regards, 

on behalf of

Dr. Osama Farouk 

Academic Editor

PLOS ONE